# Arthralgia Induced by BRAF Inhibitor Therapy in Melanoma Patients

**DOI:** 10.3390/cancers12103004

**Published:** 2020-10-16

**Authors:** Martin Salzmann, Karolina Benesova, Kristina Buder-Bakhaya, Dimitrios Papamichail, Antonia Dimitrakopoulou-Strauss, Hanns-Martin Lorenz, Alexander H. Enk, Jessica C. Hassel

**Affiliations:** 1Department of Dermatology and National Center for Tumor Diseases, University Hospital Heidelberg, Im Neuenheimer Feld 460, 69120 Heidelberg, Germany; Martin.Salzmann@med.uni-heidelberg.de (M.S.); dr.buder@gmx.de (K.B.-B.); alexander.enk@med.uni-heidelberg.de (A.H.E.); 2Division of Rheumatology, Department of Medicine V, University Hospital Heidelberg, Im Neuenheimer Feld 410, 69120 Heidelberg, Germany; karolina.benesova@med.uni-heidelberg.de (K.B.); Hannes.Lorenz@med.uni-heidelberg.de (H.-M.L.); 3Clinical Cooperation Unit Nuclear Medicine, German Cancer Research Center, Im Neuenheimer Feld 280, 69120 Heidelberg, Germany; dimitris.papamihail@gmail.com (D.P.); ads@ads-net.de (A.D.-S.)

**Keywords:** melanoma, BRAF, BRAF inhibitor, arthralgia, rheumatoid arthritis

## Abstract

**Simple Summary:**

BRAF inhibitors (BRAFi) are standard of care for BRAF-mutated metastatic melanoma (MM). One of the most common side effects is arthralgia, for which a high incidence has been described, but whose clinical presentation and management have not yet been characterized. The aim of this retrospective study was to assess the patterns and clinical course of this drug-induced joint pain and to discuss a potential pathogenesis based on our clinical findings. In our cohort of patients treated with BRAFi between 2010 and 2018, 48 of 154 (31%) patients suffered from new-onset joint pain, which primarily affected small joints with a symmetrical pattern, as can be observed in patients affected by rheumatoid arthritis, the most frequent rheumatic and musculoskeletal disease. Most cases were sufficiently treated by non-steroidal anti-inflammatory drugs; however, some patients required dose reduction or permanent discontinuation of the BRAFi. Interestingly, we found that the occurrence of arthralgia was associated with better tumor control.

**Abstract:**

Introduction: BRAF inhibitors (BRAFi), commonly used in BRAF-mutated metastatic melanoma (MM) treatment, frequently cause arthralgia. Although this is one of the most common side effects, it has not been characterized yet. Methods: We retrospectively included all patients treated with BRAFi +/− MEK inhibitors (MEKi) for MM at the National Center for Tumor Diseases (Heidelberg) between 2010 and 2018 and reviewed patient charts for the occurrence and management of arthralgia. The evaluation was supplemented by an analysis of frozen sera. Results: We included 154 patients (63% males); 31% (48/154) of them reported arthralgia with a median onset of 21 days after the start of the therapy. Arthralgia mostly affected small joints (27/36, 75%) and less frequently large joints (19/36, 53%). The most commonly affected joints were in fingers (19/36, 53%), wrists (16/36, 44%), and knees (12/36, 33%). In 67% (24/36) of the patients, arthralgia occurred with a symmetrical polyarthritis, mainly of small joints, resembling the pattern typically observed in patients affected by rheumatoid arthritis (RA), for which a role of the MAPK signaling pathway was previously described. Patients were negative for antinuclear antibodies, anti-citrullinated protein antibodies, and rheumatoid factor; arthritis was visible in 10 of 13 available PET–CT scans. The development of arthralgia was linked to better progression-free survival and overall survival. Conclusion: Arthralgia is a common side effect in patients receiving BRAFi +/− MEKi therapy and often presents a clinical pattern similar to that observed in RA patients. Its occurrence was associated with longer-lasting tumor control.

## 1. Introduction

BRAF inhibitors (BRAFi), which are usually used in combination with MEK inhibitors (MEKi), are important in the therapy of BRAF-mutated metastatic melanoma (MM). Arthralgia is acknowledged as one of the most common adverse events (AE) of BRAFi therapy. However, dedicated research on its clinical presentation, relevance, pathogenesis, and management is currently not available.

As shown in Table 1, up to about two-thirds of patients treated with vemurafenib monotherapy experience arthralgia of any grade; however, severe arthralgia (common terminology criteria for adverse events (CTC-AE) grade 3–4 [1]) was reported in less than 3–7% of patients [2,3,4,5,6,7,8,9,10]. In contrast, lower incidences of arthralgia were reported for dabrafenib (25–35% of patients) [11,12,13,14] and for encorafenib monotherapy (44% of patients) [9]. Only few publications report the incidence of arthralgia in patients treated with a combination therapy of BRAF and MEK inhibitors [9,12,14,15]. An analysis of real-world patients revealed arthralgia or myalgia in 32.9% of patients, with higher rates in those receiving BRAFi monotherapy [16]. Since rheumatologic AEs are frequently seen in patients treated with various anticancer medications, the CTC-AE v5.0 have now been adapted for their proper evaluation, after previous versions likely underestimated the impact of this side effect on patients’ activities of daily living [17,18].

Collectively, arthralgia under BRAFi is commonly seen for all mono- and combination treatments as one of the most prevalent, class-specific side effects. However, no study has ever described its clinical presentation and relevance in patient care, with the exception of some case reports [19,20,21]. Furthermore, no study has been investigating its pathogenesis.

The aim of our analysis was to better describe the clinical patterns of arthralgia and its management during treatment with BRAFi and propose mechanisms of pathogenesis based on the clinical findings.

## 2. Methods

### 2.1. Patients

We systematically included all patients treated with BRAFi with or without MEKi in the approved doses for MM at the National Center for Tumor diseases (NCT) at the University Hospital of Heidelberg, Germany, between 01/2010 and 10/2018 and retrospectively screened clinical data for the incidence and localization of arthralgia and its management. Data were collected during clinical routine, no data were collected for the purpose of this study. Since the study was designed to describe a class-specific side effect occurring with both monotherapy and combination therapy [22], patients receiving monotherapy were included, even though this is no longer the treatment standard. All conclusions on progression-free survival (PFS) and overall survival (OS) were backed by multivariate analyses due to potential bias of patients receiving monotherapy.

The patient selection process is displayed in Figure 1. Final follow-up was completed on 31 January 2019. Patients administered concurrent immune checkpoint blockers and patients not evaluable due to external treatment monitoring or duration of therapy of less than 15 days were excluded. Patients with known joint disease before the start of the therapy or arthralgia due to metastatic disease of the joint region were removed from further analysis of the arthralgia group. The BRAFi encorafenib in combination with the MEKi binimetinib has been approved for clinical use in Germany in 2018; due to the different power of this drug combination and its peculiar affinity during binding to melanoma cells, the only patient treated with it was excluded from the analysis.

### 2.2. Data Collection

Clinical and laboratory checks were routinely performed before treatment start, 2 weeks and 4 weeks after start, and then every 4 weeks. Imaging was performed every 12 weeks using CT of the neck/chest/abdomen or whole-body FDG-PET/CT scan, each combined with an MRI of the brain. Response to treatment was defined according to the Response Evaluation Criteria in Solid Tumors (RECIST) version 1.1 [23] and the PET Response Evaluation Criteria for Immunotherapy (PERCIMT) [24]. FDG-PET/CT was not done regularly, but if available, PET-CT scans were retrospectively evaluated for radiologic signs of arthritis, indicated by a newly emerging or enhanced diffuse periarticular radiotracer uptake in the joints.

We retrospectively analyzed frozen blood samples of patients for laboratory parameters of inflammation, including anti-citrullinated protein antibodies (ACPA), antinuclear antibodies (ANA), and rheumatoid factor (RF). C-reactive protein (CRP) levels were assessed regularly, and the value at the date of onset of arthralgia retrospectively documented. All patients with available samples had given their consent to retrospective research projects at the time of their blood draw.

### 2.3. Statistical Analysis

Statistical analysis was performed using IBM SPSS Statistics, version 27. PFS and OS were calculated from initiation of BRAFi treatment until progression or death from any cause, respectively. In patients with no events of progression or death at the time of the final data analysis, the date of last contact was used for censored calculation. Survival was estimated by the Kaplan–Meier method. Univariate comparisons of Kaplan–Meier estimators were performed using the log-rank test. Two-sided Fisher’s exact and Chi-square tests were used for comparisons among groups with categorical variables; *p* values were considered significant if *p* < 0.05.

### 2.4. Ethical Approval

Retrospective analyses of clinical data were approved by the institutional review board of the Medical Faculty of the University Hospital Heidelberg (no. S-069/2010). The ethical committee had agreed to the retrospective analysis of routinely collected clinical data without prior informed consent of patients.

## 3. Results

### 3.1. Patient Characteristics

We included 154 patients, 63% (97/154) male and 37% female, with a median age of 56 years (range 21–86 years). The identified BRAF mutation was V600E in 128 patients (83%), V600K in 15 (10%), V600R in 2 (1%), and V600G in 1 patient (1%); the type of V600-mutation was unknown in 8 patients (5%). Eighty-five patients (55%) were treated with vemurafenib monotherapy, 4 patients (3%) with vemurafenib + cobimetinib, 13 patients (8%) with dabrafenib monotherapy, and 52 patients (34%) with dabrafenib + trametinib. Overall, 98 patients (64%) were treated with BRAFi monotherapy, and 56 (36%) with combination therapy of BRAFi and MEKi. The median duration of therapy was 4.2 months (0.5–93.5). In addition, 88 patients (57%) received no previous systemic treatment for MM, 44 (29%) patients were administered one previous line of therapy, and 22 (14%) two or more previous lines, with a maximum of four. Forty-five patients (29%) were previously treated with immune checkpoint inhibitor (ICI) therapy, 22 of which received PD1-directed therapy as monotherapy or in combination with ipilimumab. The last line of ICI treatment was discontinued due to progression in 36 patients (80%) and due to unacceptable toxicity in 9 patients (20%); 44 patients (29%) were administered previous adjuvant interferon therapy. The response rate (RR) of our cohort was 55% (79/154), and the disease control rate (DCR) 75% (109/154). Median PFS was 5.3 months, median OS was 10.1 months; further results on treatment efficacy are described in the respective sections. All 154 patients had adverse events due to BRAFi treatment. In 8 cases (5%), therapy was ongoing at the time of data collection. Patient characteristics are shown in Table 2.

### 3.2. Arthralgia

Arthralgia occurred in 48 of the 154 patients, with an incidence of 31.2% in our cohort. The median onset of arthralgia was at 21 days (range 1–338), and the median duration was 65 days (range 2–2623). We found that 26 patients (54.2%) reported spontaneous cessation of arthralgia despite ongoing therapy, and 21 patients (43.8%) had symptoms until discontinuation; for one patient, the exact date of arthralgia cessation is unknown. No patient reported symptoms beyond the discontinuation of BRAFi therapy.

The localization of arthralgia was documented in detail in 36 of the 48 patients; 7 patients reported generalized arthralgia, and in 5 cases, the localization was unknown. Overall, small joints were predominantly affected (27/36, 75%); less patients reported involvement of large joints (19/36, 53%). Arthralgia affected most frequently the finger joints (19/36, 53%) and wrists (16/36, 44%), followed by knees (12/36, 33%), ankles (10/36, 28%), shoulders (8/36, 25%), and feet/toes (6/36, 17%). Hips or back were involved in one case each. In the majority of patients, arthralgia occurred symmetrically (32/41, 78%, 7 not evaluable). All patients with the involvement of small joints presented arthralgia in several small joints. A summary of the presentation of arthralgia is shown in Table 3.

Cases and data were critically discussed and compared to those of known diseases. With symmetrical polyarticular involvement of primarily small joints, the pattern of involvement resembled the typical presentation of rheumatoid arthritis (RA) in 24 of 36 cases (67%). As the regular rheumatic conditions’ symptom of morning stiffness was not documented thoroughly in clinical routine and not included in the analysis, the 1987 American College of Rheumatology (ACR) classification criteria for rheumatoid arthritis were not applicable [25]. Only 19% of patients (7/36) fulfilled the 2010 European League Against Rheumatism (EULAR)/ACR classification criteria [26]. However, the exact number of involved joints may be underestimated in clinical routine documentation without rheumatologic examination, and serology did not provide points in the scoring system.

### 3.3. Influencing Factors in the Development of Arthralgia

We found that 35% (34/98) of patients treated with BRAFi monotherapy developed arthralgia, compared to 25% (14/56) of those undergoing treatment with BRAFi + MEKi. This difference in the occurrence of arthralgia between patients treated with BRAFi monotherapy and those receiving combination treatment was not statistically significant in our cohort (*p* = 0.254). Considering all studied agents, there was a trend towards the highest incidence of arthralgia in patients treated with vemurafenib (36% (30/84 patients) vs. 26% (18/70 patients), *p* = 0.182). Patients treated with ICI before initiation of the BRAFi treatment had significantly lower rates of de novo arthralgia (38% (41/109) vs. 16% (7/45), *p* = 0.009). However, five of these patients required prednisolone treatment at a dose of 5–80 mg per day for immune-related adverse events (irAE), and none of these patients developed arthralgia during BRAFi treatment. Of four patients with previous PD1 inhibitor therapy-induced arthralgia, only one developed arthralgia also during BRAFi treatment, which additionally affected small joints. The development of arthralgia was also suppressed by dexamethasone treatment for brain metastases, with arthralgia documented in only 4 of 28 patients (14%, vs. 35% (44/126), *p* = 0.033). One patient developed arthralgia after discontinuing dexamethasone 7 weeks into BRAFi treatment.

### 3.4. Imaging

For 13 of 48 symptomatic patients, FDG-PET/CT scans during BRAFi treatment were available. Imaging confirmed arthritis in 77% (10/13) of patients, which corresponded to the clinically described joint region of arthralgia in 7 of them (58%). Eight patients (62%) had also increased FDG uptake in asymptomatic joints. Figure 2 shows an example of enhanced FDG uptake in the joints of a symptomatic 70-year-old patient treated with dabrafenib and trametinib. Data on visible arthritis of asymptomatic patients were not collected.

### 3.5. Serology

The levels of CRP, as an unspecific inflammatory parameter, were elevated in 79% of the cases (34/43). Confronted with symptoms mimicking rheumatic diseases, and RA in particular, we analyzed the sera of symptomatic patients for typical serological markers of RA. This included ACPA, RF, and ANA titers. Frozen sera were available for 36 of the 48 patients (79%) in the arthralgia cohort. ACPA were not detected in any patient, RF was elevated (30.1 IU/mL) in one patient, ANA were significantly elevated in one patient (1:10,000) with an unspecific speckled pattern. Slightly elevated ANA titers of 1:320 to 1:1280 were found in seven patients, with unspecific fluorescence patterns and without positive extractable nuclear antigen antibody (ENA) titers or further clinical signs of connective tissue disease.

### 3.6. Management of Arthralgia

Fourteen of the 48 patients (38%) did not require or refused any treatment for arthralgia. The majority of patients were sufficiently managed with only metamizole sodium or non-steroidal anti-inflammatory drugs (NSAIDs) like etoricoxib, indomethacin, diclofenac, or ibuprofen (22 of 48 patients, 46%).

In 6 of 48 patients (13%) low dose steroids (≤5 mg prednisone equivalent) were added to metamizole sodium or NSAIDs. In 6 of 48 patients (13%) with refractory arthralgia, dose reduction of the BRAFi was necessary in the course of treatment, in 2 cases BRAFi therapy had to be discontinued.

### 3.7. Treatment Efficacy

The median PFS was 5.3 months (95% confidence interval (CI) 4.7–5.9). This regarded patients treated by BRAFi monotherapy as well as combination therapy. For patients receiving combination treatment, the median PFS was 5.4 months (95% CI 4.6–6.2). An overview of factors influencing PFS is shown in Table 4; all of them were subjected to multivariate Cox regression due to the potential bias of different treatment regimens and disease status. PFS was significantly higher in the arthralgia cohort (median 7.9 months (95% CI 5.7–10.1) vs. 4.2 months (95% CI 3.3–5.1 months), *p* = 0.001, Figure 3), as confirmed by multivariate analysis. Patients requiring dose reduction or steroid treatment for arthralgia did not have a shortened PFS compared to patients with no treatment or NSAIDs only (*p* = 0.950). Also, glucocorticoid intake (5.7 months vs. 3.2 months, *p* = 0.000) and elevated lactate dehydrogenase (LDH) levels (6.3 months vs. 3.6 months, *p* = 0.000) had an impact on PFS. Median OS was 10.1 months (95% CI 8.8–11.4) and significantly higher in the arthralgia cohort (14.9 months (95% CI 12.4–17.4) vs. 8.7 months (95% CI 7.8–9.6), *p* = 0.006). For OS, an influence of arthralgia (odds ratio 0.62 (95% CI 0.40–0.96), *p* = 0.033), elevated LDH (odds ratio 2.59 (95% CI 1.76–3.68), *p* < 0.001), and concurrent steroid treatment (odds ratio 1.65 (95% CI 1.04–2.62), *p* = 0.033) was found in multivariate analysis.

## 4. Discussion

Arthralgia induced by BRAFi therapy has not been characterized in detail previously, despite its clinical relevance. Our results suggest a pattern with primarily symmetrical involvement of small joints as observed in patients affected by RA, the most common rheumatic disease in adults. However, only a minority of patients fulfilled the 2010 EULAR/ACR classification criteria of RA, with routine clinical documentation likely underestimating the number of joints involved when no rheumatologic assessment was performed. Similarly, for rheumatic immune-related adverse events induced by checkpoint inhibitors, only 20% of patients fulfilled the respective criteria [27]. In contrast to irAEs, no patient showed other symptoms of rheumatic disease beyond arthralgia, such as sicca or vasculitis. Unspecific inflammatory parameters were found in 79% of patients with BRAFi-induced arthralgia, as found for almost three-quarters of ICI-treated patients with irAEs showing elevated CRP levels. Specific markers of rheumatic and connective tissue diseases were not detected in symptomatic patients, as is usually the case for irAEs [17,18,27]. This is of particular interest, as seropositivity for RF and ACPA is closely linked to a more aggressive course of disease and radiographic signs of bone damage [28]. The negativity of serological markers suggests that preformed autoantibodies for a rheumatic condition are not a prerequisite to develop the respective symptoms. However, a significant proportion of RA patients also remains seronegative throughout the course of the disease. Although new autoantibodies continue to be discovered in RA and are not yet used in clinical routine [29], the self-limiting course of BRAFi arthralgia supports a pro-inflammatory potential of BRAFi different from the pathogenic mechanisms of autoimmune diseases.

Previous research into the role of the MAPK signaling pathway has proposed BRAF abnormalities as a potential pathomechanism in RA: Weisbart et al. reported the BRAF V600R mutation in synovial fibroblasts in two of nine RA patients [30] and aberrant BRAF splice variants in six of nine RA patients [31]. Interestingly, mutated BRAF was detected more frequently in the peripheral blood lymphocytes of RA patients than in controls [32] and was independent of treatment modalities. BRAF was also found to be a target of autoantibodies in RA patients [33,34], but in contrast to RF and ACPA, these are not screened for in clinical routine. An established testing kit was not available for analysis in sera obtained from our patients for further investigation of this interesting link.

Thus, a possible hypothesis for arthralgia during BRAFi treatment is a paradoxical activation of the MAPK pathway in synovial tissue and lymphocytes by BRAF V600E inhibition, with other MAPK activating mutations possibly taking place. This effect might be comparable to the well-described mechanism of development of cutaneous epithelial tumors during BRAFi treatment caused by cutaneous RAS mutations [35,36]. Relevant MAPK pathway mutations will likely include additional mutations to BRAF V600R, which is supported by the fact that in a reported case, Babacan et al. did not find this mutation in their patient [21]. The hypothesis of paradoxical MAPK activation would be in accordance with the early onset of the joint symptoms and their quick resolution after discontinuation of the therapy. The idea of targeting the MAPK signaling pathway as a predominant cause of inflammation in RA was already successfully tested by Thiel et al. [37] and later by Yamaguchi et al. [38] but has not yet brought to a novel treatment option. Recently, a case of ICI-induced polymyalgia rheumatica controlled by the MEK inhibitor cobimetinib showed the in vivo anti-inflammatory potential of targeting the MAPK signaling pathway [39]. Our data show that symptoms similar to those of RA can be triggered by interfering with or potentially activating the MAPK signaling pathway, once again supporting the idea of inhibiting MEK as a potential treatment. We would like to clearly point out that this is a hypothesis generated from clinical observations described in this study. A direct association with RA cannot be proven based on our clinical data; however, it may be of value for future research by oncologists and rheumatologists alike.

Another hypothesis is the development of an immune-related adverse event leading to arthritis due to autoreactivity similar to the side effects of ICI, as described by Ben-Betzalel et al. [40]. An indicator could be the longer-lasting tumor response in patients suffering from arthralgia, as the authors linked the development of irAEs in patients receiving BRAFi therapy with tumor control. The same was found for arthritis induced by ICI [17]. However, the clinical presentation of arthralgia is different, as ICI-induced arthritis had a median onset of 100 days and predominantly affected large joints. Furthermore, in contrast to ICI-induced irAEs, no patient developed a persisting rheumatic condition, as arthralgia in BRAFi patients was self-limiting after drug discontinuation [41]. Collectively, the clinical difference between ICI- and BRAFi-induced arthritis suggests a difference in pathogenesis. In this context, the joint symptoms can be interpreted as a surrogate parameter for enhanced proinflammatory activity that also leads to improved tumor control.

Although our study was not explicitly designed to test the diagnostic performance of FDG PET/CT in BRAFi-induced arthritis and only few cases were available, we note the relatively high incidence (10/13) of positive molecular imaging findings in symptomatic patients. Interestingly, 8 of 13 of the patients studied with PET/CT demonstrated radiologic signs of arthritis without corresponding clinical symptoms in the respective joints. These findings are in line with recent results of our group in 16 BRAF mutation-positive, metastatic melanoma patients receiving a combination therapy of vemurafenib and ipilimumab [42]. Seven patients of that cohort developed radiologic signs of irAEs, four of which presenting arthritis without clinical symptoms; importantly, these patients had a significantly longer PFS than those without radiologic irAEs, highlighting a potential relation between the appearance of irAEs signs in PET/CT and the clinical benefit of the treatment [24].

There is currently no consensus on the management of arthralgia. Welsh et al. proposed a therapy algorithm for different grades of arthralgia [43], and a recommendation is available for the management of rheumatic irAEs, including arthralgia/arthritis [18]. A standard approach to treatment is the addition of NSAID to BRAFi therapy without dose reduction for low-grade arthralgia, which was sufficient for many of our studied patients. If NSAID are not sufficient, either low-dose steroids can be added or dose reduction of BRAFi treatment can be performed. Based on our experiences and data, we propose to avoid dose reduction or discontinuation in grade 2 and, if possible, in grade 3 patients, as most cases are sufficiently treated with NSAID or, in severe cases, with concurrent low-dose glucocorticoids, which can be reduced over the course of treatment. There are no comparison data available on how treatment efficacy of BRAFi is affected by either dose reduction or addition of low-dose glucocorticoids to define the preferred way of AE management in refractory cases. Of interest, no patient received disease-modifying antirheumatic drugs (DMARDs) such as methotrexate or infliximab, which are regularly needed for adequate management of ICI-induced irAEs [18]. However, in this retrospective cohort, none of the patients was examined by a rheumatologist to discuss an indication for DMARD treatment. Possibly, the initiation of DMARDs, that may enable BRAFi continuation despite this side effect, may be of advantage for tumor control. Especially in adjuvant treatment, risks and benefits should be considered and discussed critically with the patient and a rheumatologist, as arthralgia is usually not life-threatening but can result in considerable impairment in activities of daily living and reduced quality of life.

We are aware of the multiple limitations of our study. These mainly include its retrospective nature, the limited number of patients studied, and the lack of documentation during clinical routine, leading to a reduced number of cases studied in detail (of 48 patients with arthralgia, detailed documentation on its localization was available for only 36 patients, 25% were lost). Patients were treated by clinical standards, which led to a heterogeneous distribution of the used drugs, as well as to the majority of patients receiving BRAFi monotherapy. Also, rheumatological examination was not performed collectively. Therefore, a prospective study has been launched to systematically examine and document all patients with rheumatological side effects of novel systemic treatments at our institution [44].

## 5. Conclusions

Arthralgia is a common side effect of BRAFi treatment and mainly affects small joints. Patients are typically seronegative for immunological markers of rheumatic disease; however, they present with a pattern resembling RA. A longer-lasting tumor control was observed in patients experiencing arthralgia under BRAFi treatment. Symptoms can usually be controlled by symptomatic treatment with NSAIDs or low-dose steroids. To further address the mechanisms underlying arthralgia in BRAFi-treated patients, a prospective study to characterize arthralgia clinically and by imaging (ultrasound, MRI), as well as by rheumatologic assessment, is under way.

## Figures and Tables

**Figure 1 cancers-12-03004-f001:**
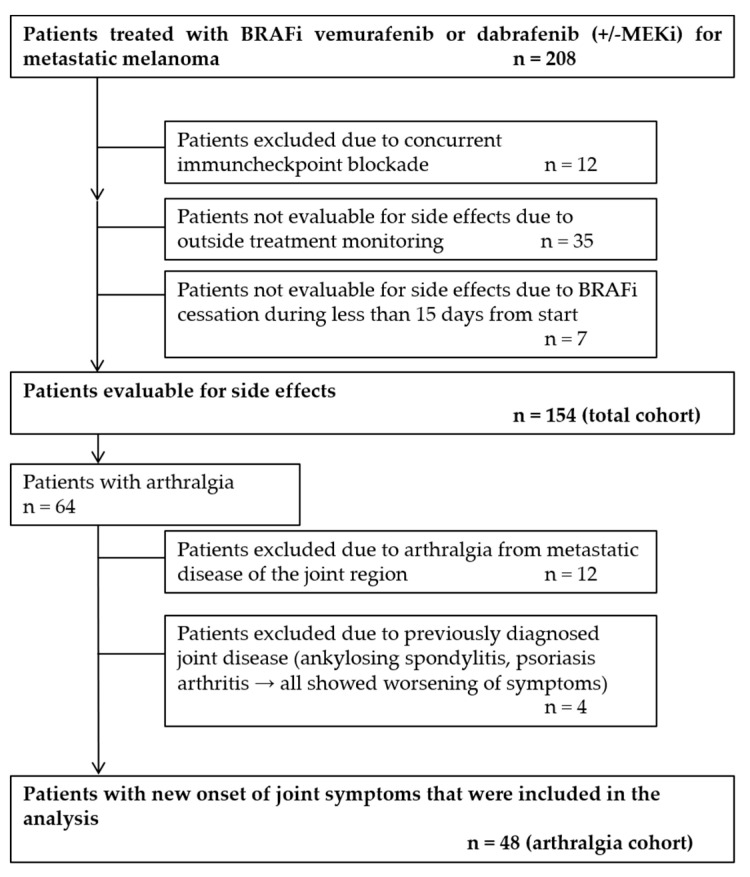
Flow diagram illustrating the process of patient selection. Cessation of BRAFi therapy in non-evaluable patients was due to unacceptable toxicity or performed at patient’s wish.

**Figure 2 cancers-12-03004-f002:**
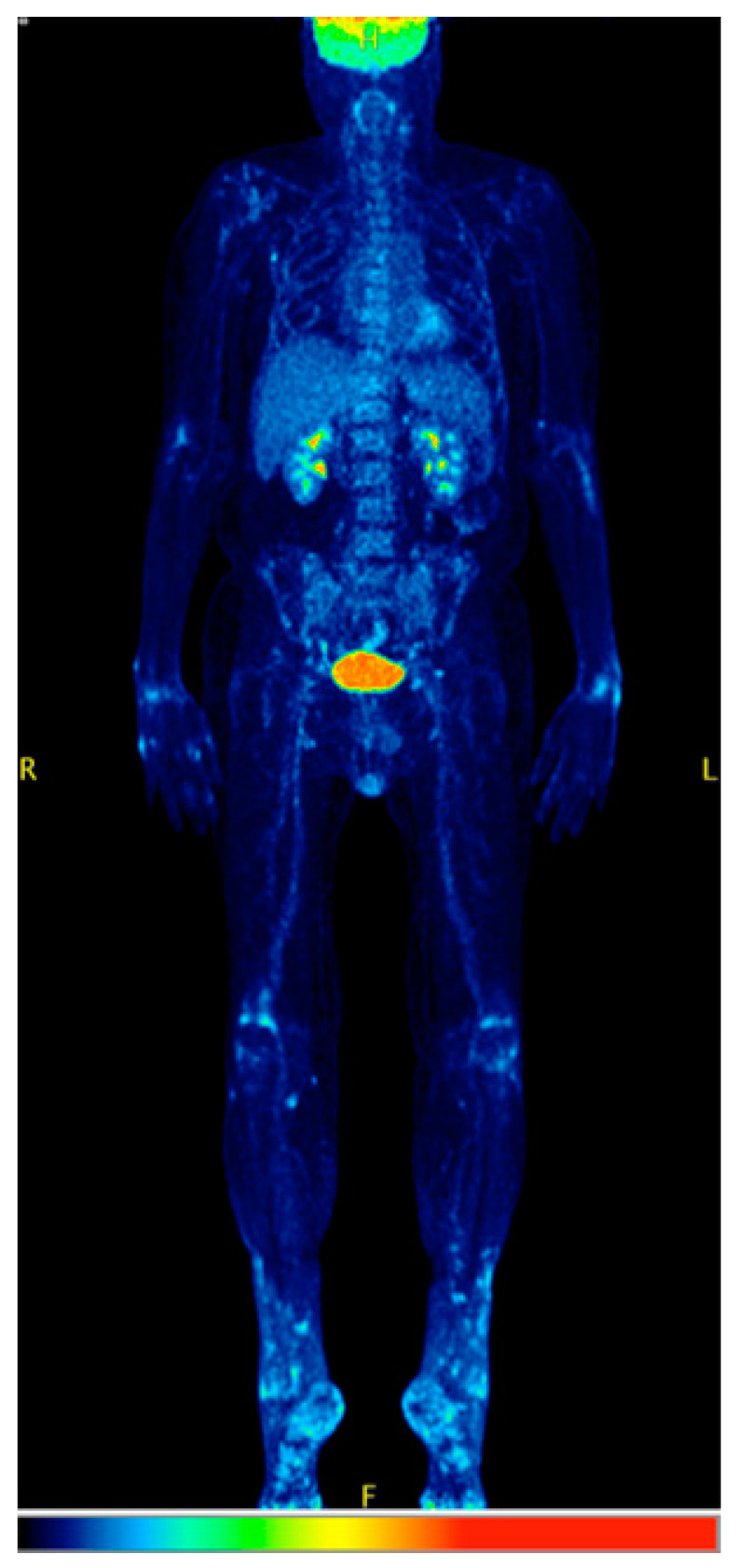
A 70-year old patient after 3 months of therapy with dabrafenib + trametinib, demonstrating radiologic signs of arthritis in PET/CT, reflected by a mostly symmetrical FDG uptake in elbows, wrists, fingers, knees, ankles, and feet (of note, the patient was in remission extracranially). The patient reported pain and swelling of fingers, wrists, and ankles.

**Figure 3 cancers-12-03004-f003:**
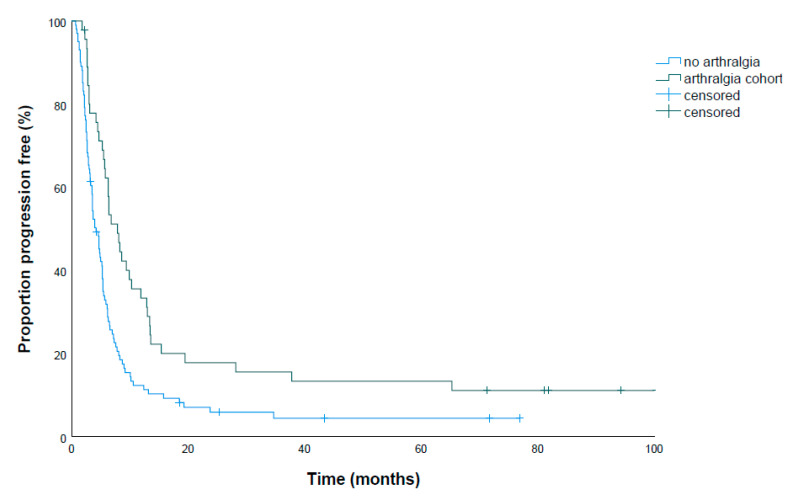
Kaplan–Meier estimated PFS shows improved PFS in patients with arthralgia (median 7.9 months (95% CI 5.7–10.1) vs. 4.2 months (95% CI 3.3–5.1 months), *p* = 0.001 (log rank)).

**Table 1 cancers-12-03004-t001:** Reported incidence of arthralgia in major publications on BRAF inhibitors (BRAFi) therapy.

Drug	Study	n	Arthralgia Any Grade (%)	Arthralgia Grade 3/4 (%)	Comments
Vemurafenib	Chapman et al. 2011 [2]	336	n/a	11 (3)	grade 1 not documented; 60 patients (18%) experienced grade 2 arthralgia
	Sosman et al. 2012 [3]	132	78 (59)	8 (6)	
	Larkin et al. 2014 [4]	3222	1259 (39)	106 (3)	
	Kim et al. 2014 [5]	336	180 (54)	15 (5)	vs. dacarbacine (3% overall)
	Arance et al. 2016 [6]	301	134 (44)	16 (5)	
	Blank et al. 2017 [7]	3219	1201 (37)	102 (3)	
	Maio et al. 2018 [8]	247	151 (61)	17 (7)	adjuvant; vs. placebo (22% overall)
	Dummer et al. 2018 [9]	186	85 (46)	11 (6)	
	Si et al. 2018 [10]	46	30 (65)	0 (0)	
Vemurafenib + Cobimetinib	Ribas et al. 2014 [15]	63	30 (48)	7 (11)	
	(Ribas et al. 2014 [15])	66	8 (12)	1 (2)	patients had progressed under previous vemurafenib therapy
Dabrafenib	Hauschild et al. 2012 [11]	100	n/a	1 (1)	Grade 1 not documented; 9 (5%) patients experienced grade 2 arthralgia
	Flaherty et al. 2012 [12]	53	18 (34)	0 (0)	
	Ascierto et al. 2013 [13]	92	30 (33)	1 (1)	
	Long et al. 2014 [14]	211	58 (27)	0 (0)	
Dabrafenib + Trametinib	Flaherty et al. 2012 [12]	55	15 (27)	0 (0)	44% any grade in 1 mg trametinib group
	Long et al. 2014 [14]	209	52 (24)	1 (0)	
Encorafenib	Dummer et al. 2018 [9]	192	85 (44)	18 (9)	
Encorafenib + Binimetinib	Dummer et al. 2018 [9]	192	54 (28)	2 (1)	

**Table 2 cancers-12-03004-t002:** Patient characteristics. MEKi, MEK inhibitors.

Parameter	Number of Patients in Total Cohort		Number of Patients in Arthralgia Cohort	
		(%)		(%)
Total number of patients	154	100	48	31.2
Age in years, median (range)	56 (21–86)		53 (28–79)	
Gender				
Male	97	63.0	28	58.3
Female	57	37.0	20	41.7
BRAF mutation				
V600E	128	83.1	42	89.4
V600K	15	9.7	4	8.5
V600R	2	1.3	0	0
V600G	1	0.6	0	0
unknown	8	5.2	2	4.2
Type of BRAFi used				
Vemurafenib	85	55.2	31	64.6
Vemurafenib + Cobimetinib	4	2.6	0	0
Dabrafenib	13	8.4	3	6.3
Dabrafenib + Trametinib	52	33.8	14	29.2
BRAFi monotherapy	98	63.6	34	70.8
BRAFi + MEKi	56	36.4	14	29.2
Duration of BRAFi treatment, median in months, (range in months)	4.2	(0.5–93.5)	7.5	(1.4–93.5)
Number of prior treatments (range)	(0–4)		(0–4)	
0	88	57.1	32	66.7
1	44	28.6	9	18.8
>1	22	14.3	7	14.6
Previous checkpoint inhibitor therapy	45	29.2	7	14.6
Previous PD1-inhibitor	22	14.2	3	6.3
Previous ipilimumab	38	24.7	5	10.4
Previous ipilimumab + nivolumab	10	6.5	1	2.1
Discontinued due to progression	36 (of 45)	80.0	6 (of 7)	85.7
Discontinued due to toxicity	9 (of 45)	20.0	1 (of 7)	14.3
Previous adjuvant interferon	44	28.6	13	27.1
Response	79	54.5	31	64.6
Disease control	109	75.2	41	85.4
Progression-free survival (median in months) [95% CI]	5.3	[4.7–5.9]	7.9	[5.7–10.1]
Overall survival (median in months) [95% CI]	10.1	[8.8–11.4]	14.9	[12.4–17.4]
Any adverse event	154	100	48	100
Treatment discontinued	146	94.8	45	93.8

**Table 3 cancers-12-03004-t003:** Clinical presentation of arthralgia.

Parameter	Number of Patients	
		(%)
Arthralgia with detailed information	36	100
Involvement of small joints	27	75
Finger joints	19	52.8
Wrists	16	44.4
Feet/toes	6	16.7
Involvement of large joints	19	52.8
Knees	12	33.3
Ankles	10	27.8
Shoulders	8	25
Hips	1	2.8
Back	1	2.8
Involvement of small joints only	17	47.2
Involvement of large joints only	9	25.0
Involvement of small and large joints	10	27.8
Symmetrical involvement	32 (of 41 evaluable)	78.0
Symmetrical polyarthritis of primarily small joints	24	66.7

**Table 4 cancers-12-03004-t004:** Influence of age, sex, tumor stage, previous immune checkpoint inhibitor (ICI) treatment, glucocorticoid intake, use of MEKi, BRAF mutation, elevated lactate dehydrogenase (LDH), and arthralgia on progression-free survival (PFS). PFS was available for 147 patients. Multivariate analysis was performed by Cox regression. Bold print indicates significant *p* values. * Two patients had advanced locoregional disease without distant metastases.

Parameter	Category	n	Univariate Analysis			Multivariate Analysis		
			Median PFS (Months)	CI (95%) (Months)	*p*	Odds Ratio	CI (95%)	*p*
**Age**	<60	89	5.3	4.0–6.6	0.603	1.03	0.71–1.48	0.881
	>60	58	4.8	4.1–5.5				
**Sex**	Male	91	4.8	3.7–5.9	0.114	0.82	0.56–1.20	0.302
	Female	56	6.2	4.9–7.5				
**Stage (brain metastases)**	M0 *–M1c	86	6.2	5.4–7.0	**0.000**	1.63	1.10–2.45	**0.017**
	M1d	61	4.2	3.0–5.4				
**Previous ICI treatment**	no	104	5.3	4.0–6.6	0.948	0.74	0.49–1.13	0.487
	yes	43	5.3	4.6–6.0				
**Glucocorticoid intake**	no	111	5.7	4.9–6.5	**0.000**	1.92	1.23–3.00	**0.004**
	yes (any dose)	36	3.2	2.6–3.8				
**MEK inhibitor used**	no	93	4.8	3.2–6.4	0.367	0.76	0.52–1.12	0.165
	yes	54	5.4	4.6–6.2				
**BRAF mutation**	V600E	122	5.4	4.6–6.2	**0.002**	1.93	1.20–3.11	**0.007**
	other	25	3.1	2.5–3.7				
**Elevated LDH**	no	74	6.3	4.8–7.8	**0.000**	2.09	1.44–3.02	**0.000**
	yes	69	3.6	2.8–4.4				
**Arthralgia**	no	101	4.2	3.3–5.1	**0.001**	0.52	0.34–0.78	**0.002**
	yes	46	7.9	5.7–10.1				

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
