# Peer review of "Arthralgia Induced by BRAF Inhibitor Therapy in Melanoma Patients"

_cancers, 2020, doi:10.3390/cancers12103004_

Round 1

Reviewer 1 Report

    This paper intent to characterize arthralgia induced by BRAFi used to treat MM via a retrospective study. The purpose is to provide clinical relevance and potential pathogenesis of the AEs. The authors presented a summarized incidence of arthralgia on BRAFi from published data in table 1, which is poorly organized not by single or combination treatment, not by year, not by prevalence high to low or any order. The text and the table did not match. The cited number ( e.g., 35%) in ref 4 is nowhere to be found in the table. The 32. 9% in ref 10 is not in the table either. I don’t think the authors make a clear case to convince the readers that there is a gap in knowledge between single BRAFi treatment and arthralgia. I don’t think they make a clear case to convince the reader that there is a gap in knowledge between single BRAFi treatment and arthralgia.
    The flow chart of the sample collection of the retrospective study in Figure 1 indicated only 48 patients with newly onset of join symptoms were selected from 154 MM patients with single or double (BRAFi+Meki) treatment. Ther are some discussions, in line 133-138, about the potential interaction on the impact of arthralgia from dual treatment and the two types of BRAFi, but the data is not clear.
    The patient characteristics are presented in Table 2, which can be improved with a clean-cut between each parameter. Especially there are typos on the number of BRAFi monotherapy and BRAFi+ Meki (according to the text, Line 108-109, it should be 98 and 56, respectively). It is even not friendly for the readers when some of the results mentioned in the text, from Line 110 to 117, can not be found in Table 2. On the other hand, some of the results, such as BRAF mutation types, were not mentioned in the text.
    All the descriptive data related to arthralgia are poorly presented. From Line 126-132, it can be assembled into a table from the 36 patients. Some of the patients have multiple types of joint pain and are repeatedly count into a different category, such as the finger, wrist, or small joints, large joints. The readers can not tell what is a significant one, except for a bunch of data with no point.
    Also, the information related to BRAFi single, double, with ICI or not were pile up in the text from Line 133-144 with minimum data support. The readers need to find each number from Table 2 if they are lucky enough to see the matched one. It is not clear and seems to hide the lack of clear comparison in study design.
    For patients FDG-PET/CT scans, based on the description from Line 83-89 of the method section, it gave the readers the impression that all 154 patients have the data. However, in Line 146, only 13 of 48 symptomatic patients FDG-PET/CT scans during BRAFi treatment were available. It is only 8% ( 13/154) of all cases and less than 30% (13/48)of arthralgia cases. It is not suitable to say in the abstract, Line 23-24, “The evaluation was supplemented by analysis of frozen sera and available FDG PET-CT scans.” The worst part is the sera data did not provide any information.
    Also, Fig 2 seems to show off the fancy image but assuming the readers know what to see. There is no legend or any information provided. And from the descriptions, line 147-149, it does not in concert with the representative Fig2.
    The data on sera, Line 155-160, does not provide any reason for using these targets for testing or explaining why it failed. The authors just presented the data and let the readers digest themselves.

    For the data on Management of arthralgia, line 162-169, it seems only the last 6 patients need to adjust BRAFi, treatment-related Management. Different pain killers can manage the rest of the cases. Arthralgia does not seem to be an issue here.
    For discussion, in Line 195, there is no mention of RA until now. The association of small joint pain to RA is a quick jump with no data provided in the data. Also, in Line 215-216, the association rooted from joint pain, RA to MAPK signaling, is stretched. This paper only has some arthralgia; pain killers can manage most of the cases. It does not associate with RA directly. From Line 227-236, there are only 13 cases of FDG PET/CT; the authors use 77% (10/13), or 62% (8/13) are misleading.
    The conclusion of BRAFi and the small joints pain is similar to other publications. The longer-lasting tumor control in patients with arthralgia under BRAFi is an interesting observation. The rest of the correlation with RA and further characterizing arthralgia with imaging ( ultrasound and MRI) may not be needed or supported by current findings.
    This paper needs a major clean up with all the numbers and re-write the association and conclusion on RA. The mechanism of longer PFS can be explored in the future.

Author Response

Dear Editors, dear Reviewer,

Thank you for your careful reading and detailed comments on our manuscript “Arthralgia Induced by BRAF Inhibitors in Melanoma”, submitted to Cancers. We apologize it seems that we did not present many of our findings clearly, and we would like to thank you for the possibility to provide a revised version. Your comments were addressed in the following manner, with changes marked in the revised document:

This paper intent to characterize arthralgia induced by BRAFi used to treat MM via a retrospective study. The purpose is to provide clinical relevance and potential pathogenesis of the AEs.

Point 1: The authors presented a summarized incidence of arthralgia on BRAFi from published data in table 1, which is poorly organized not by single or combination treatment, not by year, not by prevalence high to low or any order. The text and the table did not match. The cited number ( e.g., 35%) in ref 4 is nowhere to be found in the table. The 32. 9% in ref 10 is not in the table either. I don’t think the authors make a clear case to convince the readers that there is a gap in knowledge between single BRAFi treatment and arthralgia. I don’t think they make a clear case to convince the reader that there is a gap in knowledge between single BRAFi treatment and arthralgia.

Response 1: Table 1 has been updated thoroughly. Treatment regimens remained in the same order, listed by date of approval by the FDA (vemurafenib à dabrafenib à encorafenib), followed by the respective combination treatment. Publications were organized chronologically. Mistakes that led to inconsistency between text and table were corrected: e.g., a wrongful addition of different grades for ref. 4 (now Ref. 9, line 62), and a wrongful citation (previously ref. 10, now Ref. 16, line 65). Consistency between table and text was double checked. Citations were updated to match the order. Also, the issue of the complete lack of data on clinical presentation of the side effect was further addressed in the introduction, to better put the relevance of our retrospective data into perspective (lines 55-56, lines 73-76). Differences between the different agents were pointed out by new formulations (lines 58-62).

Point 2: The flow chart of the sample collection of the retrospective study in Figure 1 indicated only 48 patients with newly onset of join symptoms were selected from 154 MM patients with single or double (BRAFi+Meki) treatment. There are some discussions, in line 133-138, about the potential interaction on the impact of arthralgia from dual treatment and the two types of BRAFi, but the data is not clear.

Response 2: The mentioned section (now line 179-185) was updated, as the structure of the sentence did not clearly point out the findings. The key point of this section was the assessment of a (statistically non significant) difference between mono and combination treatment, which is now more clearly described. The frequent occurrence in vemurafenib monotherapy is another, separate finding. The entire section was given a new sub-header (Influencing factors on the development of arthralgia, line 179) for easier interpretation by the reader.

Point 3: The patient characteristics are presented in Table 2, which can be improved with a clean-cut between each parameter. Especially there are typos on the number of BRAFi monotherapy and BRAFi+ Meki (according to the text, Line 108-109, it should be 98 and 56, respectively). It is even not friendly for the readers when some of the results mentioned in the text, from Line 110 to 117, can not be found in Table 2. On the other hand, some of the results, such as BRAF mutation types, were not mentioned in the text.

Response 3: Table 2 was also thoroughly updated. Clean-cuts has been added for simplification between all categories. Typos were corrected and once again, all numbers double checked for consistency.

Both text and table were updated, in a way that all information of the text appears in the table in the same order (respective section “patient characteristics”, lines 132-150). Further data on previous ICI treatment was provided to this section at the request of a different reviewer (lines 142-144).

Point 4: All the descriptive data related to arthralgia are poorly presented. From Line 126-132, it can be assembled into a table from the 36 patients. Some of the patients have multiple types of joint pain and are repeatedly count into a different category, such as the finger, wrist, or small joints, large joints. The readers can not tell what is a significant one, except for a bunch of data with no point.

Response 4: Table 3 has been added to the manuscript, which features the information on arthralgia condensed in a table. Also, information on the involvement of small joints or large joints only, as well as the information if the presentation was a symmetrical polyarthritis of primarily small joints (typical for RA), were added. Unfortunately, the design of the study cannot distinguish which of the reported localizations is the clinically more significant one, as this was not documented in clinical routine.

The section has been updated to show data on ACR/EULAR classification criteria (lines 170-178) to further put the data on RA into perspective, see also Response 13.

Point 5: Also, the information related to BRAFi single, double, with ICI or not were pile up in the text from Line 133-144 with minimum data support. The readers need to find each number from Table 2 if they are lucky enough to see the matched one. It is not clear and seems to hide the lack of clear comparison in study design.

Response 5: As mentioned in Response 3, all the information provided on influencing factors were given an own section for better display of the results (see headline “Influencing factors on the development of arthralgia”, line 179). If not already in the text, all the raw data were added to each statement, so that a jump to Table 2 is not necessary (section now: lines 180-195). Table 2 was intended to provide patient characteristics of both the total cohort, and also the arthralgia cohort in the respective section, and should not be needed for data display of statements in line 180-195. Please note that the study design was not intended to compare different groups, these findings are gathered from retrospectively collected data which was intended to provide a better understanding of the toxicity.

Point 6: For patients FDG-PET/CT scans, based on the description from Line 83-89 of the method section, it gave the readers the impression that all 154 patients have the data.

Response 6: It was pointed out that PET/CT is not a standard procedure, and that the evaluation of signs of arthritis was only done if available (line 108).

Point 7: However, in Line 146, only 13 of 48 symptomatic patients FDG-PET/CT scans during BRAFi treatment were available. It is only 8% ( 13/154) of all cases and less than 30% (13/48)of arthralgia cases. It is not suitable to say in the abstract, Line 23-24, “The evaluation was supplemented by analysis of frozen sera and available FDG PET-CT scans.”

Response 7: The respective statement in the abstract was removed (lines 34-35). Also, the number of 77% positive PET scans was removed from the abstract (line 44), the absolute number (10 of 13) was kept, as it shows the low power of the analysis.

Point 8: The worst part is the sera data did not provide any information.

Response 8: We would like to respectfully disagree, as a negative result on commonly used serological markers for rheumatic diseases is useful for clinical differentiation when confronted with the symptoms. Please find this issue further addressed in Response 11.

Point 9: Also, Fig 2 seems to show off the fancy image but assuming the readers know what to see. There is no legend or any information provided.

Response 9: Please find a description of the image attached to the figure: “Figure 2: A 70-year old patient after 3 months of therapy with dabrafenib + trametinib demonstrating radiologic signs of arthritis in PET/CT, reflected by a mostly symmetrical FDG uptake in elbows, wrists, fingers, knees, ankles and feet (of note, the patient is in remission extracranially). The patient reported pain and swelling of fingers, wrists and ankles.” (lines 204-207).

Point 10: And from the descriptions, line 147-149, it does not in concert with the representative Fig2.

Response 10: The reference to Figure 2 was removed from the sub-header and pointed out that the image is an example of visible arthritis (lines 200-201).

Point 11: The data on sera, Line 155-160, does not provide any reason for using these targets for testing or explaining why it failed. The authors just presented the data and let the readers digest themselves.

Response 11: An introduction was added in lines 210-212, stating that the used markers are the most common in rheumatic diseases and in RA especially. A discussion on the findings was added in lines 262-274 within the thoroughly updated discussion. CRP levels were included as unspecific signs of inflammation (line 209, discussion lines 260-262).

Point 12: For the data on Management of arthralgia, line 162-169, it seems only the last 6 patients need to adjust BRAFi, treatment-related Management. Different pain killers can manage the rest of the cases. Arthralgia does not seem to be an issue here.

Response 12: We would like to respectfully disagree on this point as well. The data also means that the majority of patients required medical treatment for these symptoms, by pain killers / anti-inflammatory medication or even immunosuppressive treatment. Having severe joint pain requiring pain killers is an impactful issue on daily activity and health related quality of life.

Discontinuing BRAFi or reducing the dose may literally cost these patients several months of their life, so it is only done in very severe, treatment-refractory cases. Arthralgia is a side effect that required 4% (6/154) of patients to reduce the dose of a life-prolonging treatment, and 2 of 154 cases to even permanently discontinue it. This is a clinically relevant issue in daily patient contact.

An updated discussion on other potential treatments to avoid discontinuation and treatment discontinuation was added in lines 340-347.

Point 13: For discussion, in Line 195, there is no mention of RA until now. The association of small joint pain to RA is a quick jump with no data provided in the data. Also, in Line 215-216, the association rooted from joint pain, RA to MAPK signaling, is stretched. This paper only has some arthralgia; pain killers can manage most of the cases. It does not associate with RA directly.

Response 13: The clinical resemblance was added and main findings that led to the resemblance updated. This includes the addition of ACR/EULAR criteria in the results section (lines 170-178), as well as the number of patients presenting with symmetrical polyarthritis of primarily small joints as the typical presentation in RA (lines 170-172, see also Table 3). At this point, the resemblance is easier for the reader to compare to the provided data. The discussion was vastly changed to further address this resemblance in combination with the discussion of serologic data (lines 252-274). We also pointed out more clearly that the possible association with MAPK signaling is a hypothesis based on our clinical findings. The research design is not capable of proving a direct association, but is supposed to provide a clinical basis for further research both in oncology, and in rheumatology. This was emphasized in the discussion (lines 301-304).

Point 14: From Line 227-236, there are only 13 cases of FDG PET/CT; the authors use 77% (10/13), or 62% (8/13) are misleading.

Response 14: The percentages were changed to raw numbers to show the limited amout of available data (lines 320-321).

Point 15: The conclusion of BRAFi and the small joints pain is similar to other publications.

Response 15: No previous publication other than case reports (References 19-21) has reported the finding of BRAFi primarily affecting small joints. To the best of our knowledge, this is the first study to evaluate the side effect systematically and describe the involvement of small joints.

Point 16: The longer-lasting tumor control in patients with arthralgia under BRAFi is an interesting observation.

Response 16: The point was added to the conclusion section of the abstract (lines 46-47).

Point 17: The rest of the correlation with RA and further characterizing arthralgia with imaging (ultrasound and MRI) may not be needed or supported by current findings.

Response 17: We understand this issue to summarize previous points; adjustments were made when the correlation with RA was described, as well as the clinical relevance stressed, as pointed out in previous responses. We believe this association may be of great interest for oncologists when treating various tumors with a BRAF mutation, as well as rheumatologists when assessing underlying mechanisms, with the interesting link of clinical symptoms. Therefore, the prospective evaluation was launched.

Point 18: This paper needs a major clean up with all the numbers and re-write the association and conclusion on RA. The mechanism of longer PFS can be explored in the future.

Response 18: All wrongful numbers were corrected and double checked. The association and conclusion on RA was changed vastly, also pointing out that it is merely a clinical observation, and a definitive association cannot be drawn based on clinical data, as described in previous responses. We would like to keep the mechanism of longer PFS as one of the key findings.

Once again, we apologize for any inconvenience and unclear presentation. However, we would like to emphasize that arthralgia is a relevant issue when treating patients with BRAFi and that this is the first systematic retrospective analysis on this as far as we are aware of.

Hence, we do think the topic will be interesting for your readers and thank you, once again, for the possibility to provide a revision.

Sincerely, on behalf of all authors.

Prof. Dr. Jessica Hassel       Dr. Martin Salzmann

Reviewer 2 Report

Salzmann et al. retrospectively investigated 154 patients treated with BRAF+/- MEK inhibitors for metastatic melanoma and reviewed patient charts for the occurrence and management of arthralgia. They concluded the development of arthralgia by BRAF+/- MEK inhibitors correlated with better PFS and OS. This manuscript is interesting for dermato-oncologist to predict the efficacy of BRAF+/- MEK inhibitors for metastatic melanoma.

I have several minor concerns,

Since 30% of patients pre-treated with immune checkpoints inhibitors, the more information about first line therapy is needed. Please present an information about the reason to switch pre-immunotherapy (e.g. progress disease, adverse events)?

Please describe concise protocol of immunotherapy. How many patients administered nivolumab plus ipilimumab combination therapy, anti-PD1 Abs monotherapy or ipilimumab monotherapy?   

The authors could not detect the specific marker of arthralgia. If the patient’s serum is still available, serum levels of several RA-related chemokines, such as CXCL5, might be useful for this study.

Author Response

Dear Editors, dear Reviewer,

Thank you for careful reading and your kind comments on our manuscript “Arthralgia Induced by BRAF Inhibitors in Melanoma”, submitted to Cancers. We would like to thank you for the possibility to provide a revised version. Your comments were addressed in the following manner, with changes marked in the revised document:

Salzmann et al. retrospectively investigated 154 patients treated with BRAF+/- MEK inhibitors for metastatic melanoma and reviewed patient charts for the occurrence and management of arthralgia. They concluded the development of arthralgia by BRAF+/- MEK inhibitors correlated with better PFS and OS. This manuscript is interesting for dermato-oncologist to predict the efficacy of BRAF+/- MEK inhibitors for metastatic melanoma.

I have several minor concerns,

Point 1: Since 30% of patients pre-treated with immune checkpoints inhibitors, the more information about first line therapy is needed. Please present an information about the reason to switch pre-immunotherapy (e.g. progress disease, adverse events)? Please describe concise protocol of immunotherapy. How many patients administered nivolumab plus ipilimumab combination therapy, anti-PD1 Abs monotherapy or ipilimumab monotherapy?  

Response 1: Further information on previous ICI treatment lines have been added to the section “patient characteristics” (lines 142-144). 80% of patients discontinued the most recent line of ICI treatment due to progression, 20% due to unacceptable toxicity. The numbers of previous types of ICI treatment were added to Table 2 (previous PD1 in 22 cases, previous ipilimumab in 38 cases [please note: sequential treatment was possible] and dual ICI treatment in 10 cases).

Point 2: The authors could not detect the specific marker of arthralgia. If the patient’s serum is still available, serum levels of several RA-related chemokines, such as CXCL5, might be useful for this study.

Response 2: Unfortunately, we were unable to perform further analyses, as only in a small amount of patients enough sera were left. Also, we do not have a nearby cooperating laboratory to measure the requested chemokines. We did, however, add CRP levels as an unspecific marker of inflammation to the section “serology” (lines 209-210) and also added a thorough discussion on serologic findings, putting them into perspective (lines 252-274). This includes a statement on novel autoantibodies currently not used in clinical routine.

We apologize for any inconvenience and unclear presentation. Thank you, once again, for the possibility to provide a revision.

Sincerely, on behalf of all authors.

Prof. Dr. Jessica Hassel    Dr. Martin Salzmann

This manuscript is a resubmission of an earlier submission. The following is a list of the peer review reports and author responses from that submission.

Round 1

Reviewer 1 Report

Thank you for addressing all concerns.

Reviewer 2 Report

The manuscript describes data on the development of arthralgia in melanoma patients receiving combo targeted. 

Major pointss of the paper concern the heterogeneous number of patients receiving single or combo targeted therapy. Novel brafi as ecorafenib and binimetinib show different power and peculiar affinity during the binding to melanoma cells. Thus, more patients need to be excluded or this combination excluded.

The standard of care for metastatic melanoma is combo targeted. Thus, conclusions based on single agents are not appropriate and not in line with standard management of the metastatic disease. 

Authors describe this adverse event in their population but the strategy for overcoming symptoms, apart from CS, is not clear.